# OpenReview forum: "Human-Prior Correction: Scalable Post-hoc Calibration that Aligns Vision Models with Human Uncertainty"
_ICLR.cc/2026/Conference — ICLR 2026 Conference Desk Rejected Submission_

### Official Review · Reviewer_bvL3 · 2025-10-31

**Soundness:** 2
**Presentation:** 2
**Contribution:** 3
**Rating:** 2
**Confidence:** 4

**Summary:**

This paper proposes Human-Prior Correction (HPC), a post-hoc calibration method that incorporates human confusion patterns into model predictions. The key innovation is using foundation models (CLIP, DINO) as proxy sources for human-like confusions, eliminating expensive annotation requirements. The method is evaluated on CIFAR-10/100 and ImageNet variants, showing improvements in human alignment (NLL_human) and calibration (ECE) while maintaining accuracy.

**Strengths:**

- Using foundation models (CLIP, DINO) as proxy sources for human confusion patterns is creative and addresses a genuine practical bottleneck—expensive human annotations.
- The paper considers multiple important aspects of calibration (human alignment, robustness, fairness, conformal prediction), but the results are poorly organized and inconsistently reported, making it hard to extract clear conclusions from the main text.

**Weaknesses:**

**Errors in references**

- Non-existent reference: "Correcting the calibration: A novel framework for aligning classifier predictions with human judgement"
- Wrong authors: Galil et al. (2023) citation lists "Itay Galil, Shiran Dori-Hacohen, and Alon Shachaf" but actual authors are "Pedro Sandoval-Segura, Vasu Singla, Jonas Geiping, Micah Goldblum, Tom Goldstein"

These errors raise serious concerns about the paper's rigor.

**Major writing and presentation issues**

- Lines 254-260 (Ablations section): States results but doesn't reference any table or figure—where should readers verify these claimed observations?
- Lines 372-377: Discusses results without referencing specific figures or tables
- Table 4: Caption appears both above and below the table
- Table 5: Uses completely different format compared to other tables
- Table 2: Doesn't mention which model/architecture is used
- Figure 3 caption: Mentions "BCTS, Dirichlet" but these methods are not shown in the figure
- Table captions don't include main conclusions or takeaways

These issues make verification of claims impossible.

**Inconsistent reporting**

The arbitrary selection of which proxy results to show suggests cherry-picking and does not instill confidence.
Table 1: Reports Human, DINO, CLIP
Table 2: Reports Human, CLIP (why not DINO?)
Table 3: Reports CLIP, SimCLR for CIFAR-100;
CLIP, DINO for ImageNet-200

**LLM usage is not reported**

**Questions:**

Why do different tables report different subsets of proxy priors?

---

### Official Review · Reviewer_2gf5 · 2025-11-03

**Soundness:** 2
**Presentation:** 2
**Contribution:** 2
**Rating:** 2
**Confidence:** 4

**Summary:**

The paper proposes a confidence calibration method for classification networks. The calibration is done by combining the image prediction distribution with the marginal class confusion distribution of the predicted class.   The combining is done by applying the posterior formula.

**Strengths:**

Confidence calibration is an important problem and there is always room for improvement.

**Weaknesses:**

The paper is not well written. The abstract is too technical.   Notation is not defined before it is used (e.g. p_0  in eq. 4). What is beta in section 3. 4.?  Adding an algorithm box to the paper can increase its readability.

The temperature scaling method is very simple, robust and works well. The proposed method requires setting many parameters (class conditional distributions and lambda)  and there is no significant improvement that justifies using the proposed method.

**Questions:**

I didn't find the paper Li & Zhang (2025). I guess it hasn't been published yet.
What is beta in section 3.4?

Why not use the class confusion matrix computed on the calibration dataset instead of the clip-based distribution?

In the case that the predicted class y_pred is wrong, you are using the wrong class conditional distribution C[y_pred,:]. How does this affect your method?

The method is closely related to performing temperature scaling, where temperature is selected for each predicted class separately.
You should compare your method with this simple version.

**Details Of Ethics Concerns:**

I didn't find the paper Li & Zhang (2025). I guess it hasn't been published yet.

---

### Official Review · Reviewer_4sGm · 2025-11-05

**Soundness:** 2
**Presentation:** 2
**Contribution:** 2
**Rating:** 2
**Confidence:** 4

**Summary:**

The paper introduces Human-Prior Correction (HPC), a post-hoc calibration framework that adjusts model confidence to align with human perceptual uncertainty without retraining. The key idea is to blend model predictions with a human confusion prior through a Bayesian objective, yielding a closed-form correction. This prior encodes structured human confusion patterns, such as cat↔dog similarities, which are either derived from annotation datasets like CIFAR-10H or approximated using proxy priors from foundation models (CLIP, DINO, SimCLR). The paper claims HPC improves human alignment (NLL_\text{human}) by about 20%, reduces Expected Calibration Error by nearly 24%, and enhances robustness under distribution shift while maintaining accuracy. The method incurs negligible computational overhead and complements existing techniques like temperature scaling.

**Strengths:**

The proposed approach to infuse human confusion information is well-motivated. To scalably define the prior, they approximate human confusion through models like CLIP and SSL.  They conducted experiments across multiple datasets and demonstrate consistent gains across architectures, robustness to prior misspecification. The integration of HPC with conformal prediction is also well-motivated, offering tighter yet semantically coherent prediction sets.
The method’s practicality without requiring  retraining, and adding a small computational overhead is interesting.

**Weaknesses:**

While the method is presented as post-hoc, it depends on access to a class-conditional confusion prior that itself must be estimated, either from human annotations or proxies derived from other models. The assumption that CLIP or DINO capture human-like confusion patterns is plausible but not theoretically justified. While theoretical justification might be difficult, atleast there should be a discussion on when this assumption fails. For example, consider a fine-grained classification which is a practically relevant classification tasks. Say CUB-200. Wouldn't the human confusion be much larger than the CLIP/DINO when they are trained on it. Consequently, given the mismatch, the result would not align with human-aware calibration. Highlighting this is important.
Related to this, if a FM like CLIP is assumed to be available, the paper should discuss the reasons behind utilizing a Resnet-18 or Resnet-50 instead of directly deploying CLIP (with/ without finetuning) for the classification task itself.? The main novelty is primarily from approximating human-confusion with these foundation models.
Also, a lot of references are non-existent, making it very difficult to ascertain the merits as well as placing trust in the results of the paper.

Q. Li and M. Zhang. Correcting the calibration: A novel framework for aligning classifier predictions with
human judgement. Journal of Machine Learning Research, 26(5):775–800, 2025. doi: 10.5555/12345678.
12345679. 1, 2, 4
Yifan Zhang, Qiang Li, Bolei Zhou, Andrea Vedaldi, and Philip H.S. Torr. Human-aligned uncertainty quantifi-
cation for vision transformers. In Proceedings of the IEEE/CVF Conference on Computer Vision and Pattern
Recognition (CVPR), pp. 8234–8243, 2024. 2
Hongyi Zhang, Moustapha Cisse, Yann N. Dauphin, and David Lopez-Paz. On calibration of mixture-of-experts
neural networks. In International Conference on Learning Representations (ICLR), 2020. 2

**Questions:**

Please refer to the weaknesses section

**Details Of Ethics Concerns:**

References seem to be hallucinated- possibly through an LLM.

---

### Note · Program_Chairs · 2026-01-17
**Submission Desk Rejected by Program Chairs**

The following references in this submission do not refer to real documents and/or have major errors in bibliographic information:

 Q. Li and M. Zhang. Correcting the calibration: A novel framework for aligning classifier predictions with human judgement. Journal of Machine Learning Research, 26(5):775-800, 2025. doi: 10.5555/12345678.
Zhen Wang, Xiaotian Chen, Yang Liu, Jian Tang, and Philip H.S. Torr. Calibrating vision foundation models with conformal prediction. In International Conference on Learning Representations (ICLR), 2024.